# Profiling Patients by Intensity of Nursing Care: An Operative Approach Using Machine Learning

**DOI:** 10.3390/jpm10040279

**Published:** 2020-12-14

**Authors:** Honoria Ocagli, Corrado Lanera, Giulia Lorenzoni, Ilaria Prosepe, Danila Azzolina, Sabrina Bortolotto, Lucia Stivanello, Mario Degan, Dario Gregori

**Affiliations:** 1Unit of Biostatistics, Epidemiology and Public Health, Department of Cardiac, Thoracic, Vascular Sciences, and Public Health, University of Padova, via Loredan, 18, 35121 Padova, Italy; honoria.ocagli@unipd.it (H.O.); corrado.lanera@unipd.it (C.L.); giulia.lorenzoni@unipd.it (G.L.); ilaria.prosepe@unipd.it (I.P.); danila.azzolina@uniupo.it (D.A.); sabrina.bortolotto@studenti.unipd.it (S.B.); 2Department of Translational Medicine, University of Piemonte Orientale, 28100 Novara, Italy; 3Health Professional Management Service (DPS) of the University Hospital of Padova, 35128 Padova, Italy; lucia.stivanello@aopd.veneto.it (L.S.); mario.degan@aopd.veneto.it (M.D.)

**Keywords:** Barthel index, machine learning technique, CLARA, intensity of nursing care

## Abstract

Physical function is a patient-oriented indicator and can be considered a proxy for the assignment of healthcare personnel. The study aims to create an algorithm that classifies patients into homogeneous groups according to physical function. A two-step machine-learning algorithm was applied to administrative data recorded between 2015 and 2018 at the University Hospital of Padova. A clustering-large-applications (CLARA) algorithm was used to partition patients into homogeneous groups. Then, machine learning technique (MLT) classifiers were used to categorize the doubtful records. Based on the results of the CLARA algorithm, records were divided into three groups according to the Barthel index: <45, >65, ≥45 and ≤65. The support vector machine was the MLT showing the best performance among doubtful records, reaching an accuracy of 66%. The two-step algorithm, since it splits patients into low and high resource consumption, could be a useful tool for organizing healthcare personnel allocation according to the patients’ assistance needs.

## 1. Introduction

In medicine, the importance of evaluating patient-centered outcomes in both clinical research and in the management area has grown in recent years [1]. This approach has long been advocated in the “P4 medicine” healthcare paradigm (predictive, preventive, personalized and participative) [2]. Physical function, along with quality of life, is one of the patient-centered indicators of health status [3]. In detail, physical function (PF) defines the level of activity of daily living of patients, and it decreases physiologically in the elderly population or in particular disease conditions [4]. A loss in basic functions, such as activities of daily living (ADLs), leads to a reduction in the independence of the patient. Therefore, physical function assessment is a useful indicator for proper planning and managing resources correctly, especially in elderly patients in the medium to long term [5]. It can be used as a proxy for the evaluation of autonomy in ADLs and consequently for the assignment of healthcare personnel, such as nurses and nurse assistants, in terms of both numbers and specialization. In the hospital setting, the parameter used for the assignment of personnel is the level of intensity of nursing care in medical terms. However, a high level of intensity of nursing care—which implies higher costs in terms of drugs and procedures—does not necessarily reflect high requests for assistance. The Veneto region in Italy has solved the problem of classifying patients according to the complexity of care, with deliberation no. 52 of 20 May 2014 enacting the “Piano Socio Sanitario Regionale 2012–2016” [6]. According to this regulation, the number of nurses needed for the patient’s care is determined by the calculation of the time employed to care for the patient during his or her stay at the hospital.

Other Italian regions have used similar systems. However, these systems, in several cases, do not consider the complexity of patient care. The computation of the number of nurses and nurse assistants dedicated to patient caregiving is a problem widely debated in the literature. A recent European study (RN4CAST) involving more than 25,000 nurses showed that a small number of nurses negatively influenced the mortality rate after surgical procedures [7]. In the Italian context, more specifically, the nurse-patient ratio is lower than in other European countries. An increase in this ratio could improve safety and reduce burnout episodes, especially in younger nurses [8]. Instruments reported in the literature to classify patients according to their complexity of care have been based mainly on clinical evaluation. Although these instruments have a different grade of validity for classification, they do have some limitations, especially for use related to general resource planning. In healthcare systems, there is an increasing interest in machine learning techniques (MLTs), which are useful when managing large datasets and with highly correlated variables, such as diagnosis, intervention, and physical function. MLTs have been applied especially to aid clinicians in diagnosis and predictive prognosis, but their application is expanding in all healthcare areas [9].

In the nursing literature, there remains a paucity of data-driven classification systems, especially in clinical decision support. However, in nursing fields, there is also growing application of MLTs [10], for example, in supporting decision making [11] and in health management [12]. This study aims to develop a machine learning-based model algorithm for classifying patients into homogeneous groups according to the intensity of nursing care needed, using administrative data derived from the reimbursement systems and information about physical function evaluated with a standard tool, i.e., the Barthel index (BI).

## 2. Materials and Methods

### 2.1. Study Design and Population

This study is a retrospective study of administrative data recorded between 2015 and 2018 at the University Hospital of Padova (Italy). Patients older than 18 years old admitted to the hospital in an ordinary regime (no day-hospital activity) with a length of stay of at least three days (the minimum in-hospital length of stay for having the BI completed at discharge) were included in the study. Patients admitted to intensive care and to the maternal unit were excluded.

### 2.2. The Barthel Index Scale

The Barthel index is an instrument widely used worldwide for the evaluation of patients’ ability to perform basic activities of daily living (ADLs) [13]. The tool evaluates the following abilities: feeding, bathing, grooming, dressing, bowel and bladder control, toilet use, transfer (bed to chair and back) mobility, and stair climbing. Items can be scored between 0 (not able at all) and 15 (independent). The total score can range from 0, which implies total dependence, to 100, representing a patient who does not require any help with ADLs. The instrument has shown good performance in various populations, and it has been recently used for evaluating patient outcomes, especially in elderly patients [14]. In Veneto, the compilation of the Barthel instrument is mandatory in each hospital, and it is included in the discharge documentation; however, it is not included in the reimbursement system, which is based on the diagnosis-related groups (DRG) system.

### 2.3. Ethics

This study is based on administrative data without any involvement of the patients. Patients admitted to our hospital provided their consent to use their data for research papers. The study was approved by an internal review board. Data were treated anonymously according to the current regulations, both Italian and European.

### 2.4. Statistical Analysis

Descriptive analysis is reported as quartiles I, II (median), and III for continuous variables and as percentages and absolute numbers for categorical variables.

### 2.5. Machine Learning Classification

An MLT algorithm was used to cluster patients according to their clinical features. The following variables were considered for the computation: age, gender, admission date, the first diagnosis of discharge classified according to the International Classification of Disease (ICD-9), department of admission, and the partial score of the Barthel index. The steps of the MLT classification are shown in Figure 1.

#### 2.5.1. Cluster Analysis

A large clustering applications (CLARA) algorithm [15] was considered for clustering the patients into homogeneous groups. CLARA is a modification of the classical partitioning around medioids (PAM) clustering algorithm that is suitable for big data (N > 10,000). The number of investigated clusters spans from 2 to 5 groups. The silhouette information criterion was considered to evaluate the performance of each clustering strategy. We evaluated the overall mean silhouette width for each strategy and the distribution of the silhouettes among each cluster for all strategies (Appendix A). The cluster analysis was performed with the cluster package [16] in R software [17].

#### 2.5.2. Gold Standard Creation

Groups distinguished with the CLARA algorithm do not necessarily have clinical meaning in terms of patient care needs. Patients who are not clearly distinguished by the algorithm (i.e., having an intermediate BI level) fall into a doubtful zone. A randomly selected sample of patients located in the doubtful zone was manually classified by an expert nurse according to the groups defined by the CLARA clustering algorithm. The expert nurse used additional information to classify patients, such as integrated clinical nursing notes, Braden scale for risk of pressure ulcers [18], Conley scale for risk of falling [19], vascular access, nursing assessment sheets, and, whenever available, the nursing transfer/discharge letter. MLT algorithms were trained and tuned on the gold standard and classified into the main groups identified by the CLARA algorithm.

#### 2.5.3. Machine Learning Technique Classifiers Applied to Doubtful Cases

The MLTs used in doubtful cases were as follows: (i) random forest (RF), a supervised learning algorithm that randomly creates and merges multiple decision trees into one “forest” [20]; (ii) a generalized linear model net (GLMN), which fits a generalized linear model via penalized maximum likelihood [21]; and (iii) a support vector machine (SVM) [22], which is a classifier defined by a separating hyperplane. The two-step algorithm, given by the combination of CLARA and the MLT with the best performances, was then applied to the overall sample. The performance of each model was evaluated using the following measures: accuracy, sensitivity, specificity, area under the receiver operating characteristic (ROC) curve (AUC), and F1 score. The performance values refer to those given by the validation part of the 10-fold cross-validation (which was repeated 5 times) for the best tune for each model.

## 3. Results

The records included in the study number 74,514. Of these records, 52% (38,768) were from women with a median age of 64 years old admitted mainly in medical (30%; 22,348) and critical (28%; 20,955) areas (Table 1).

### 3.1. Machine Learning Classifiers Results

The best CLARA algorithm was the one that clustered records into two groups: low resource consumption (LRC) and high resource consumption (HRC) (Figure 1). Analyzing the subjects, we noted that, in both groups, some patients were uniquely defined from the clinical point of view: all patients with BI < 45 were clustered into the HRC group, while all patients with BI > 65 were clustered into the LRC group. However, patients with BI scores between 45 and 65 presented a greater challenge since, in this region, the clusters provided by the CLARA algorithm overlapped. For these reasons, we finally divided patients into three groups: (i) group 1, formed by patients with a BI less than 45, who represent a higher cost in terms of assistance; (ii) group 2, composed of subjects with BI greater than 65, who do not require particular assistance in their activities of daily living; and (iii) group 3, characterized by patients with BI scores between 45 and 65, who belong to a doubtful zone that must be further investigated via MLT (Appendix A).

An expert nurse manually classified 1000 records of the 6271 records belonging to the doubtful group (group 3) to create the gold standard. Among the gold standard cases, 475 records were placed in the HRC group, while 525 were placed in the LRC group.

The records classified by the expert’s opinion served as the gold standard for the training of the RF, SVM, and GLMN algorithms. Table 2 reports the performance of each best-tuned model. SVM was the technique that revealed the best performance among the three models, reaching an accuracy of 66% (95% CI 0.59–0.75), a sensitivity of 62% (95% CI 0.51–0.74), and a specificity of 72% (95% CI 0.62–0.83).

The two-step algorithm was then applied to the overall sample. In the Appendix A, Appendix A reports the predictive performances over group 1, group 2 and the gold standard. For the overall sample, the performances of the algorithm were considerably increased compared to those of the CLARA algorithm alone, with a sensitivity of 98%, a specificity of 99%, and an accuracy of 98%.

### 3.2. Descriptive Results According to the Two-Step Algorithm

Table 3 reports the characteristics of the sample according to the classification of the two-step algorithm. Patients classified in the HRC group were mainly admitted to medical wards (42%; 429,436) and older than 65 years old (37%; 3,713,528). Partial BI was lower in patients classified in the HRC group not able to feed themselves (96%; 5553), with no bowel control (95%; 97,235), with no bladder control (91%; 9,110,140), dependent in bathing (60%; 6,015,914), unable to climb stairs (74%; 7,415,426), and unable to use the toilet alone (99%; 9,912,132).

The LRC group was more represented than the HRC group in each year considered. Gender was equally distributed in each group (Table 3). The distribution of LRC and HRC among the departments was similar for the three years considered. Every year, the HRC group was mainly represented in emergency-urgency departments (68%; 77) and medical departments (41%; 4170) in 2016 (Table 4).

## 4. Discussion

This study classifies patients on the basis of administrative data scoring according to physical function evaluation. Administrative data are a rich and inexpensive source of general epidemiological information [23].

The two-step algorithm classifies patients into two main categories based on the level of care assistance needed in activities of daily living. The percentage of patients classified at a high level of assistance in the overall sample was 21.6%. The patients considered to be at a high level of assistance are prevalently elderly, are admitted to emergency urgency and medical departments, and have a lower score on the partial Barthel index. The group classified as having a low level of assistance was instead characterized by patients with higher partial BI scores. These patients were mainly women and were principally admitted to the mental health ward or surgical departments.

The problem of nursing and nursing assistant allocation is particularly relevant in healthcare systems in terms of the cost and quality of care provided. These two professional figures represent the largest segments of healthcare systems; for this reason, the management of this professional component has a relevant impact in terms of healthcare costs. Despite the high impact of nursing work in the healthcare system, this component is not considered in reimbursement systems, especially in Italian systems [24].

Other instruments for the evaluation of patients′ complexity of care have been reported in the literature. However, they cannot always be used because some of them were created for the intensive care unit setting or require further data (e.g., clinical data or other external information) for their utilization. The project research of nursing, for example, requires the creation of a detailed list of nursing activities [25]. In the Italian context, for example, the informative system of nursing performances (SIPI) considers the nurse′s opinion on the complexity of patient care, instead of the time dedicated to a single assistance activity, to properly organize the clinical staff dedicated to caregiving activities [26]. The professional assistance method (MAP) is another method used in Italy for the determination of the complexity of patient care and the identification of the correct number of healthcare providers. The MAP is a supportive method for nursing care planning, and it has the objective of proposing and testing a method for the development of standard care plans concerning specific health problems [27]. To allocate personnel in Finnish hospitals, the RAFAELA [28] criterion is applied. In China, a new classification system was recently proposed that can be used to define the number of nurses and nursing assistants. This method considers both disease severity and activities of daily living scores to classify patients [29]. In the Italian context, in contrast, SIPI has also been proposed as a system of classification for the complexity of patient care oriented toward the proper allocation of nurses [26].

This work proposes a two-step classification algorithm leading to automatic profiling of patients into high- and low-assistance groups. The first step is based on a cluster analysis aimed at identifying a specific group of patients with particular care needs. Cluster analysis is the most commonly used MLT in the nursing literature [30]. In previous studies, MLTs were used to predict high-risk patients and treat them with specific pathways [31].

The second classification step of the algorithm proposed in this work is based on MLT classification algorithms. The literature has demonstrated that these methods are useful when managing a large amount of data, as shown in research aimed at identifying critical elements for predicting risk status in nursing documentation [32]. Another recent study in the nursing field considered MLTs and administrative data to identify patients at high risk for hospital readmission [33].

In all of the examples reported, MLTs were used as support tools for clinical decisions; this work instead proposes a two-step algorithm as a support tool for organizational management.

### Limitations of the Study

This study was a single-center study considering the data provided by a university hospital. Future research developments are needed to apply and validate the two-step algorithm using similar data.

## 5. Conclusions

To our knowledge, there has been a paucity of studies in the nursing field attempting to classify patients using administrative data to make clinical decisions with the support of automatic tools. The main result of this study is that patients admitted to the hospital can be classified as having a high level or low level of assistance needed according to both medical information and physical activities. This objective is crucial in nursing research, especially when considering specific populations, such as older people, who often require more nursing and nursing assistant aid in daily care routines due to their decline in physical function.

The two-step algorithm could be used as a support tool in organizational management. In a subsequent phase, the two-step algorithm could be used to automatically profile patients as high or low assistance needed at admission. This procedure will allow us to organize the allocation of a greater number of professionals to these departments, which mainly consist of patients requiring additional care.

The early identification of patients who require more nursing assistance could improve the quality of care, reduce negative outcomes for both healthcare providers and patients and help in the planning of appropriate pathways according to the needs of the patient. The strength of this two-step algorithm, compared to other systems for quantifying the personnel needed, lies in the clusterization being data-driven and not derived from an a priori classification. Moreover, the data required in this two-step algorithm do not require specific information related to the patient, which may not be available at all times.

Thus, MLTs, when applied to high-dimensional data, offers the opportunity to classify patients requiring a high level of nursing assistance accurately. These techniques are favored over traditional statistical models since they can identify complex relations between data, improving the prediction ability. Further studies should be performed to improve model generalizability by including data from other countries. However, future work is needed to improve these results by implementing them in clinical practice.

## Figures and Tables

**Figure 1 jpm-10-00279-f001:**
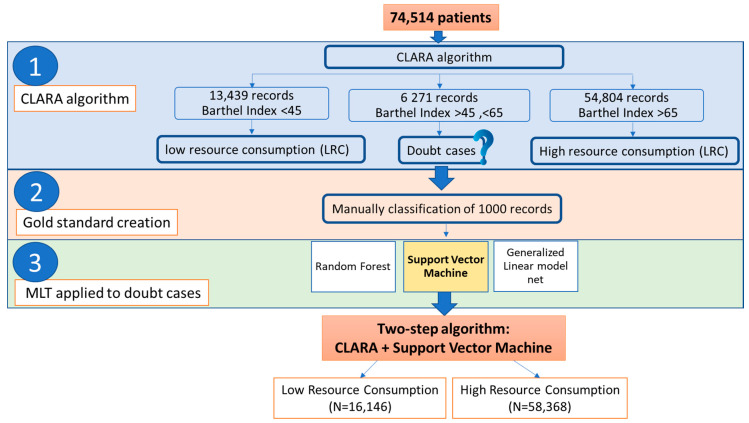
Two-step algorithm. This figure explains in detail how the machine learning technique (MLT) classification works in our two-step algorithm for the identification of low and high resource consumption.

**Table 1 jpm-10-00279-t001:** Descriptive characteristics of the sample. Continuous variables are reported as quartiles I, II (median) and III and categorical variables as percentages and absolute frequencies % (N).

Variable	Level	*N* = 74,514
Feeding	0	8% (5785)
	5	12% (9284)
	10	80% (59,445)
Bowel control	0	10% (7602)
	5	7% (5353)
	10	83% (61,559)
Bladder control	0	15% (11,097)
	5	7% (5018)
	10	78% (58,399)
Mobility	0	15% (11,384)
	5	8% (6008)
	10	14% (10,574)
	15	63% (46,548)
Bathing	0	36% (26,559)
	5	64% (47,955)
Grooming	0	19% (14,077)
	5	81% (60,437)
Stair climbing	0	28% (20,723)
	5	20% (15,009)
	10	52% (38,782)
Bed to chair transfers	0	12% (8658)
	5	10% (7508)
	10	11% (8595)
	15	67% (49,753)
Toilet use	0	16% (12,292)
	5	12% (8877)
	10	72% (53,345)
Dressing	0	14% (10,473)
	5	19% (14,207)
	10	67% (49,834)
Age		45/64/77
Age category	<65	51% (38,043)
	>65	49% (36,471)
Gender	Female	52% (38,768)
	Male	48% (35,746)
Department	S	28% (20,955)
	CTV	15% (10,807)
	W-C	13% (9903)
	E-U	0% (240)
	M	30% (22,348)
	N-SO	13% (9610)
	MH	1% (651)

S = surgical, CTV = cardio-thoraco-vascular, W-C = woman–child, E-U = emergency-urgency, M = medical, N-SO = neurosciences-sense organs, MH =mental health.

**Table 2 jpm-10-00279-t002:** Predictive performances with 95% confidence intervals of the three MLT classification models were evaluated using the following measures: accuracy, sensitivity, specificity, area under the curve (AUC), and F1 score. These values refer to those given by the validation part of the 10-fold cross-validation (which is repeated 5 times) for the best tune for each model.

	Accuracy	Sensibility	Specificity	AUC	F1
Random forest	0.62	0.42	0.81	0.69	0.51
	(0.56, 0.70)	(0.31, 0.53)	(0.72, 0.89)	(0.62, 0.78)	(0.40, 0.62)
GLMNet	0.66	0.57	0.74	0.72	0.62
	(0.56, 0.72)	(0.45, 0.67)	(0.63, 0.82)	(0.64, 0.76)	(0.50, 0.68)
Support vector machine	0.66	0.62	0.72	0.72	0.63
	(0.59, 0.75)	(0.51, 0.74)	(0.62, 0.83)	(0.63, 0.8)	(0.54, 0.73)

**Table 3 jpm-10-00279-t003:** Classification of high and low levels of resource consumption of the two-step algorithm according to the following variables: functional area, year, age, gender, and partial Barthel index. The variables are all categorical and are reported as percentages and absolute frequencies: % (N).

Variables	2-Step Algorithm	Variable Levels							Combined
**Functional area**		**S** **(*N* = 20,955)**	**CTV** **(*N* = 10,807)**	**W-C** **(*N* = 9903)**	**E-U** **(*N* = 240)**	**M** **(*N* = 22,348)**	**N-SO** **(*N* = 9610)**	**MH** **(*N* = 651)**	***N* = 74,514**
	*High*	14% (3019)	18% (1950)	1% (70)	76% (182)	42% (9436)	15% (1448)	6% (41)	22% (16,146)
	*Low*	86% (17,936)	82% (8857)	99% (9833)	24% (58)	58% (12,912)	85% (8162)	94% (610)	78% (58,368)
**Year**		**2015** (*N* = 523)	**2016** (*N* = 31,829)	**2017** (*N* = 22,322)	**2018** (*N* = 19,840)				*N* = 74,514
	*High*	48% (249)	22% (7014)	22% (4929)	20% (3954)				22% (16,146)
	*Low*	52% (274)	78% (24,815)	78% (17,393)	80% (15,886)				78% (58,368)
**Age**		**<65** (*N* = 38,043)	**>65** (*N* = 36,471)						*N* = 74,514
	*High*	7% (2618)	37% (13,528)						22% (16,146)
	*Low*	93% (35,425)	63% (22,943)						78% (58,368)
**Gender**		**female** (*N* = 38,768)	**male** (*N* = 35,746)						*N* = 74,514
	*High*	22% (8616)	21% (7530)						22% (16,146)
	*Low*	78% (30,152)	79% (28,216)						78% (58,368)
**Barthel index levels**									
**Feeding**		**0** (*N* = 5785)	**5** (*N* = 9284)	**10** (*N* = 59,445)					*N* = 74,514
	*High*	96% (5553)	76% (7046)	6% (3547)					22% (16,146)
	*Low*	4% (232)	24% (2238)	94% (55,898)					78% (58,368)
**Bowel control**		**0** (*N* = 7602)	**5** (*N* = 5353)	**10** (*N* = 61,559)					*N* = 74,514
	*High*	95% (7235)	87% (4643)	7% (4268)					22% (16,146)
	*Low*	5% (367)	13% (710)	93% (57,291)					78% (58,368)
**Bladder control**		**0** (*N* = 11,097)	**5** (*N* = 5018)	**10** (*N* = 58,399)					*N* = 74,514
	*High*	91% (10,140)	70% (3516)	4% (2490)					22% (16,146)
	*Low*	9% (957)	30% (1502)	96% (55,909)					78% (58,368)
**Walking**		**0** (*N* = 11,384)	**5** (*N* = 6008)	**10** (*N* = 10,574)	**15** (*N* = 46,548)				*N* = 74,514
	*High*	98% (11,201)	64% (3825)	10% (1093)	0% (27)				22% (16,146)
	*Low*	2% (183)	36% (2183)	90% (9481)	100% (46,521)				78% (58,368)
**Bathing**		**0** (*N* = 26,559)	**5** (*N* = 47,955)						*N* = 74,514
	*High*	60% (15,914)	0% (232)						22% (16,146)
	*Low*	40% (10,645)	100% (47,723)						78% (58,368)
**Grooming**		**0** (*N* = 14,077)	**5** (*N* = 60,437)						*N* = 74,514
	*High*	87% (12,317)	6% (3829)						22% (16,146)
	*Low*	13% (1760)	94% (56,608)						78% (58,368)
**Climbing scale**		**0** (*N* = 20,723)	**5** (*N* = 15,009)	**10** (*N* = 38,782)					*N* = 74,514
	*High*	74% (15,426)	5% (705)	0% (15)					22% (16,146)
	*Low*	26% (5297)	95% (14,304)	100% (38,767)					78% (58,368)
**Bed-to-chair transfers**		**0** (*N* = 8658)	**5** (*N* = 7508)	**10** (*N* = 8595)	**15** (*N* = 49,753)				*N* = 74,514
	*High*	100% (8628)	85% (6393)	13% (1075)	0% (50)				22% (16,146)
	*Low*	0% (30)	15% (1115)	87% (7520)	100% (49,703)				78% (58,368)
**Toilet use**		**0** (*N* = 12,292)	**5** (*N* = 8877)	**10** (*N* = 53,345)					*N* = 74,514
	*High*	99% (12,132)	44% (3894)	0% (120)					22% (16,146)
	*Low*	1% (160)	56% (4983)	100% (53,225)					78% (58,368)
**Dressing**		**0** (*N* = 10,473)	**5** (*N* = 14,207)	**10** (*N* = 49,834)					*N* = 74,514
	*High*	98% (10,269)	40% (5680)	0% (197)					22% (16,146)
	*Low*	2% (204)	60% (8527)	100% (49,637)					78% (58,368)

S = surgical, CTV = cardio-thoraco-vascular, W-C = woman–child, E-U = emergency-urgency, M = medical, N-SO = neurosciences-sense organs, MH = mental health.

**Table 4 jpm-10-00279-t004:** Algorithm classification according to department and year of admission.

		Department							Combined
	2-Step Algorithm	S	CTV	W-C	E-U	M	N-SO	MH	
2016	N	(*N* = 8552)	(*N* = 4432)	(*N* = 4376)	(*N* = 113)	(*N* = 10,244)	(*N* = 3891)	(*N* = 221)	(*N* = 31,829)
	High	16% (1334)	18% (807)	1% (29)	68% (77)	41% (4170)	15% (578)	9% (19)	22% (7014)
	Low	84% (7218)	82% (3625)	99% (4347)	32% (36)	59% (6074)	85% (3313)	91% (202)	78% (24,815)
2017	N	(*N* = 6441)	(*N* = 3320)	(*N* = 3032)	(*N* = 68)	(*N* = 6273)	(*N* = 2966)	(*N* = 222)	(*N* = 22,322)
	High	15% (950)	20% (678)	1% (22)	87% (59)	43% (2726)	16% (482)	5% (12)	22% (4929)
	Low	85% (5491)	80% (2642)	99% (3010)	13% (9)	57% (3547)	84% (2484)	95% (210)	78% (17,393)
2018	N	(*N* = 5872)	(*N* = 2967)	(*N* = 2495)	(*N* = 49)	(*N* = 5543)	(*N* = 2707)	(*N* = 207)	(*N* = 19,840)
	High	12% (701)	15% (442)	1% (19)	78% (38)	43% (2374)	14% (371)	4% (9)	20% (3954)
	Low	88% (5171)	85% (2525)	99% (2476)	22% (11)	57% (3169)	86% (2336)	96% (198)	80% (15,886)

S = surgical, CTV = cardio-thoraco-vascular, W-C = woman–child, E-U = emergency-urgency, M = medical, N-SO = neurosciences-sense organs, MH = mental health.

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
