# Peer review of "Profiling Patients by Intensity of Nursing Care: An Operative Approach Using Machine Learning"

_jpm, 2020, doi:10.3390/jpm10040279_

Round 1

Reviewer 1 Report

Would encourage the authors to add on a prospective design in which the data from ML were implemented in a management strategy for the patients from a general and not just Italian perspective.

Author Response

Dear reviewer,

thank you for your time and effort that you have dedicated to providing feedback to our work.

Introduction has been revised (lines 32, 37-40, 62-65).

Design and methods were modified (line 202). Figure 1 was added to explain step by step how the classification works with our data (line 103).

Results section has been improved by providing further details in the text and reducing table 1 (line 143).

Would encourage the authors to add on a prospective design in which the data from ML were implemented in a management strategy for the patients from a general and not just Italian perspective.

We agree with the reviewer and we have added the comment to the discussion (line 256).

Reviewer 2 Report

This is a well-written manuscript and I only have a couple of recommendations: 

Page 2, line 68 suggest describing what ordinary regimen means 

Did the researchers get IRB exempt status? 

One of the limitations is the use of one instrument to measure functional status. this was retrospective data so generalizable is limited 

suggest have  more information about future implications for research and how this applies to direct patient care/personalized medicine 

Author Response

Dear reviewer,

thank you for your time and effort that you have dedicated to providing feedback to our work.

Further details have been provided to the methods section by adding Figure 1 (line 202), which explains step by step how the classification works with our data.

This is a well-written manuscript and I only have a couple of recommendations: 

Page 2, line 68 suggest describing what ordinary regimen means 
Thanks, information has been added as requested by the reviewer.

Did the researchers get IRB exempt status? 
The study was approved by an internal review board. This information has been added to the methods section (line 92).

One of the limitations is the use of one instrument to measure functional status. this was retrospective data so generalizable is limited 

suggest have  more information about future implications for research and how this applies to direct patient care/personalized medicine 

Comments about future implication of the research work have been added to the discussion section (line 245).

Submission Date

20 October 2020

Date of this review

02 Nov 2020 19:49:42

Reviewer 3 Report

This study aims at developing a machine-learning-based model algorithm for classifying patients in homogeneous groups according to the intensity of nursing care needed using administrative data derived from the reimbursement systems and information on physical function evaluated with a standard tool, the  Barthel Index.

The objective of the paper is of crucial relevance for the research on nursing, especially in the field of geriatrics, where multimorbidity, polypharmacy and fragility constitute a burden for the health systems and for the formal caregivers. 

The methodology adopted to defined the two steps-algorithm for the identification of homogenuos groups with HCR or LRC is in line with the research trends in the health sector and it favours of the availability of huge amount of data. This represents the added value of the project for the scientific community.

The use of the Barthel Index is appropriate, but it to underline that the tool is used to assess level of autonomy not only merely physical function. In fact, the topic of physical function seems to be present only in the Introduction part. I suggest to the authors to slighty moves the attention from physical function to the independence/autonomy level, in the Introduction paragraph (lines 30.36).

The abstract should be improved as follow:

  • Pls, correct the incomplete statement in line 17-18:
  • Pls, correct line 21, the acronym of MLTs is not explained before.

Author Response

Dear reviewer,

thank you for your time and effort that you have dedicated to providing feedback to our work.

This study aims at developing a machine-learning-based model algorithm for classifying patients in homogeneous groups according to the intensity of nursing care needed using administrative data derived from the reimbursement systems and information on physical function evaluated with a standard tool, the  Barthel Index.

The objective of the paper is of crucial relevance for the research on nursing, especially in the field of geriatrics, where multimorbidity, polypharmacy and fragility constitute a burden for the health systems and for the formal caregivers. 

The methodology adopted to defined the two steps-algorithm for the identification of homogenuos groups with HCR or LRC is in line with the research trends in the health sector and it favours of the availability of huge amount of data. This represents the added value of the project for the scientific community.

We have included the potential implications of the study findings at the discussion section (line 233 and 247).

The use of the Barthel Index is appropriate, but it to underline that the tool is used to assess level of autonomy not only merely physical function. In fact, the topic of physical function seems to be present only in the Introduction part. I suggest to the authors to slighty moves the attention from physical function to the independence/autonomy level, in the Introduction paragraph (lines 30.36).

Done, we have moved the attention to independency/autonomy level (line 37).

The abstract should be improved as follow:

  • Pls, correct the incomplete statement in line 17-18:
  • Pls, correct line 21, the acronym of MLTs is not explained before.

Thanks, modified as suggested.

Submission Date

20 October 2020

Date of this review

17 Nov 2020 11:58:00